# A Review of Climate Security Discussions in Japan

**DOI:** 10.3390/ijerph19148253

**Published:** 2022-07-06

**Authors:** Christo Odeyemi, Takashi Sekiyama

**Affiliations:** Graduate School of Advanced Integrated Studies in Human Survivability, Kyoto University, Kyoto 606-8306, Japan; sekiyama.takashi.2e@kyoto-u.ac.jp

**Keywords:** climate security, systematic literature review, Japan’s climate policy, climate securitization, climate change, human security, climate crisis

## Abstract

This review paper provides preliminary analysis and answers to three key questions that were identified by synthesizing qualitative evidence from climate security research in Japan. The questions identified are: (1) Has Japan participated in the global climate security debate at all? (2) Why did climate security struggle to become a major political theme in Japan until 2020? (3) Why did Japan explicitly start dealing with climate security as a policy issue in 2020? We identify and discuss four key reasons relative to the second question. The review provides key details (and general parameters) of these questions that have been overlooked by not only Japanese researchers but also climate security research conducted between 2017 and 2022 in Europe and the United States. Climate security suddenly became a trending topic among Japanese researchers and political elites in 2020; we find evidence that future studies could provide important and more robust insight if an analysis of the above questions is supported by interview data obtained from Japanese government officials. In doing so, researchers will be able to provide valuable insight into the possibility (and extent) that inter-ministerial rivalry between key ministries has impeded domestic progress on climate security action. Furthermore, three separate projects on climate security have been commissioned and recently implemented in Japan. These form the basis for this first systematic literature review of 34 papers and the related research reports resulting from these projects. These papers and reports were retrieved from the electronic databases of Google Scholar, ProQuest, and the National Institute for Environmental Studies in April 2022. While the main limitation of this review paper is that readers are expected to connect these questions to their own experiences at the global level, we reduce the possibility of presenting biased information by identifying and verifying missing details. For example, we had difficulty identifying the method used in one of the co-authored papers and contacted the corresponding author. In summary, sustained discussion in academia and high-politics settings should eventually lead to a greater awareness about climate security.

## 1. Introduction

Both United Nations (UN) Security Council member countries and non-member countries continue to debate the global issue of climate change and security—conceptualizing the idea of climate security remains a priority among researchers in the West, where the climate security field itself is distinctly different from other areas of the debate and scientific evidence has formed the basis for conceptual, if somewhat controversial, contributions in propelling national and international action [1,2,3,4,5]. While similar goals to further conceptualize climate security are ongoing in the European Union [6,7,8] and the United States, the global debate has been led not by the academic community but by the policy community, who have guided and shaped the related climate security discourse more than academic researchers [1,2]. However, although defense policymakers and their scenario reports occupy the front row in climate security genealogy, climate security discourse is neither static nor uncontested, as the quality of discussions in the academic community has clearly improved [2].

As the level of understanding on climate change increases, so does the literature on climate security [1] and related policies. This means that the research to date has contributed considerably to the important concept of climate security, but there remain important aspects to be investigated at the national and international level. For example, Rita Floyd [3] argues that the phenomenon of ideological fragmentation that often accompanies the institutional fragmentation of global climate security governance automatically impedes the realization of climate security for people. This is because, according to Floyd, the preferences of diverse security organizations and actors often deviate significantly from the UN’s explicit preference of understanding climate security from a human security perspective. However, this is not our aim here. So, we did not seek to resolve the lack of a widely accepted definition of climate security [2,3,4], despite the fact that the traditional concept of security is closely tied to the notion of national security, and that climate change and security discourse has opened the door for non-state actors such as civil society, local governments, and the business sector to participate [4]. It is worth noting that some scholars have suggested climate security as a relevant concept in the context of climate governance and related policies. However, such perspective is not without its problems. For example, despite its focus on the effects of climate change on conflict, climate security thinking not only neglects the reality that adaptation measures can foster conflict, but often provides only minimal analyses of political processes and structures [2]. In the same vein, there have been predictions that climate change could cause political unrest and armed conflicts. These projections include an increase from 350,000 climate-related deaths today to 1 million per year by 2030, an increase from USD 100 million in economic losses to USD 3 billion, and a 10-fold increase in degradation if no action is taken (Conference Report, https://www.wiltonpark.org.uk/wp-content/uploads/wp1167-report.pdf, accessed on 26 June 2022; as cited in Selby and Hoffmann [2]). One of the two scholars at the conference, and coincidentally one of the co-authors of the paper by Selby and Hoffmann [2], objected to these gloomy predictions on the grounds that there is little scientific consensus on the conflict and security implications of climate change. The objection was immediately refuted by a speaker on the basis of the actual lack of consensus on all manner of issues [2]. Given that Japanese climate security researchers and policymakers are not unaware of similar disagreements within Japan—the world’s third-largest economy and the world’s fourth-largest provider of official development assistance—the state of climate security and its policy considerations in the country need further examination.

However, Japan has completely missed the trend of international debate on climate security, although Japanese decision-makers are aware of global trends in climate discussions because the Diet began discussing climate security as early as 2007. It is also unfortunate that climate security remains unfamiliar in Japan due to the paucity of research in this area [9,10]. This is partly due to the lack of robust policy debate on climate security in Japan, although it can also be questioned whether research really has a strong role in enabling policy discussion. For example, the terms “environmental security” or “climate security” cannot be found in the Annual Reports on Environment in Japan, published by the Ministry of the Environment (MOE), from 2007 to 2020. Similarly, a search of the Defense White Paper shows no mention of environmental security or climate security until 2020. Perhaps reflecting this lack of policy interest, climate security has not received much attention in Japanese academic circles. A search for “climate security” on CiNii Articles, a Japanese article index database provided by the National Institute of Informatics, yielded only eight hits as of February 2022; all eight were published between 2007 and 2010. This suggests that climate security has rarely been a research topic in Japanese academia, except for a few years after 2007, when the MOE compiled a report on climate security.

The purpose of our review is to identify Japan’s current climate security policy, determine directions for future research on this topic, and contribute to the climate security research base. Because understanding these topics requires clarifying how Japanese researchers have discussed the climate security framework and how political elites have debated and positioned this framework, this compelled a review of some very recent studies on climate security by Japanese researchers in order to illustrate the key details (and general parameters) of three important questions that these researchers have not identified. The general implication from our review is the need for future and in-depth examination of climate security policy in Japan. In other words, to fully answer the questions identified above, future researchers need to look at how politicians have positioned climate security from a security perspective and the arguments Japanese researchers have made relative to that positioning. This review consists of three parts. First, we outline the methodology and sampling strategy. Second, we present results for the key parameters of each question that are not covered by the latest wave of climate security research published in Japan over the last few years. Third, we discuss and conclude our findings.

## 2. Research Method

The systematic literature review method is increasingly used in the context of environmental change research [11]. Basically, it aims to answer specific research questions by systematically mapping relevant themes in the available literature on a topic, allows findings to be matched to predefined eligibility criteria, and helps the articulation of a broader and impartial overview [12,13,14]. Compared with traditional meta-analytic evaluation, a systematic review permits the thorough determination of the general aspects of a study (such as type, number and geographic focus) and is particularly useful for interdisciplinary investigation that involves quantitative and qualitative methods [15]. A systematic review is also useful in a scenario of streetlight effect. The streetlight effect refers to the tendency of researchers to focus on specific questions, cases or variables for reasons of convenience or data availability rather than wider relevance, policy importance or construct validity [16]. The present systematic review follows the five major steps enumerated in the *Cochrane Handbook for Systematic Reviews of Interventions*, which was last updated in 2022, and notes that while construct validity is largely dependent on the choice of methodology, there is no one size fits all. The steps include the identification of research questions, the identification of databases and initial search phrases, the specification of string queries (or criteria for the selection of studies to be included in the review), and a discussion of the research questions (see Appendix A: PRISMA 2020 Main Checklist, Appendix A: PRIMSA Abstract Checklist). We enumerated a distilled form of these steps as follows.

### 2.1. Identification of Research Questions

We identify important research directions by uncovering three crucial questions that are neither fully covered nor thematically analyzed by the climate security research community in Japan. For this reason, we formulate three research questions: (1) Has Japan participated in the global climate security debate at all? (2) Why did climate security struggle to become a major political theme in Japan until 2020? (3) Why did Japan explicitly start dealing with climate security as a policy issue in 2020? These questions are applicable at the global level and are key directions for future research on climate security.

### 2.2. Identification of Databases and Initial Search Phrases

The electronic databases of the National Institute for Environmental Studies (NIES, Tsukuba, Ibaraki, Japan), Google Scholar, and ProQuest were consulted. The latter two databases in particular offer a variety of publications and a set of search options. Search criteria were applied to the literature search because it was not the goal of this study to conduct an exhaustive systematic review [17]. Inclusion and exclusion criteria were also applied to significantly narrow down the number of papers. Peer-reviewed articles, review articles, book chapters and conference proceedings were included in the search, while non-peer-reviewed publications and abstracts were excluded. Besides ensuring that the research was up to date, the decisive reason for conducting a literature search focusing on papers published between 2017 and 2022 was that the global climate debate experienced significant progress around 2019, with a new dimension of public debate, especially in Europe and the United States.

### 2.3. Specification of the String Query

A string query was applied to the NIES, Google Scholar, and ProQuest databases. Specifically, the following string query was used: climate security discourse AND Japan AND policy debate. This query helped to extract papers that were highly relevant to the three research questions and to search for works by researchers who focused on the case of Japan, with climate security as the object of investigation (Figure 1).

## 3. Results

Given that the term climate security became common in Japan around 2020 [1], we have to emphasize not only the relevance of the Japanese policy debate and the academic research used as primary source material for the present review, but also Olivia Lazard’s [6] assertion that, since 2007—when the UN Security Council first discussed the topic of climate security—policy and research have focused on adding climate components to existing approaches to fragility and conflict. Therefore, we foreground two recent research projects on climate security in Japan. While such research is important because climate security policies must focus not merely on adapting to unknown and higher levels of unpredictability, but on fostering the changes necessary to rebuild ecological balance and stability at the global level [8], these projects are particularly relevant for understanding climate security in Japan because they reveal what gaps and discrepancies exist between academia and policy.

A preliminary search of the NIES, Google Scholar, and ProQuest databases for the phrase “climate security debate” yielded research papers and reports based on two research projects that were conducted in Japan. It should also be noted that the number of hits from the preliminary search was too large to conduct a fruitful review. For example, the search on NIES yielded 153 results (https://www.nies.go.jp/index-e.html, accessed on 15 March 2022) while Google Scholar returned 1,420,000 hits (https://bit.ly/3rwQsAm, accessed on 15 March 2022). After applying the string query (“climate security” discourse AND Japan AND policy debate), the total number of papers from the three databases was reduced to 602 English language papers, with 49 papers from ProQuest, 545 papers from Google Scholar (including duplicates from ProQuest, https://bit.ly/3uLfWMe, accessed on 15 March 2022), and 8 papers from NIES (Google search in the upper right corner, https://www.nies.go.jp/index-e.html, accessed on 15 March 2022). Unlike the Google Scholar and ProQuest databases, the NIES database does not have the option to apply all search criteria. The titles and abstracts of all 602 papers were scanned to ascertain clear relevance to the goal of our review. Whenever we are unable to make this determination in a single paper, the full text of the paper was scanned (or read when necessary) for relevance to the three research questions. For the evaluation of eligibility and relevance, we narrowed the number of papers to 34 (by specifically including the word Japan): 2 from NIES, 10 from ProQuest, 15 from Google Scholar, and 7 reports from a research project.

The purpose of the first research project (titled “Study on socioeconomic risks to Japan caused by global climate change impacts”; https://bit.ly/3MgQu7j, accessed on 15 March 2022) was to understand the term compound risk as it applies to climate change and what it means for Japan’s security. Co-funded by MOE and the Environmental Restoration and Conservation Agency, the project was conducted from 2018 to 2021 with the participation of 16 researchers, mostly Japanese. According to Yasuko Kameyama [10] (the project leader), this project categorized climate risks into six types (or themes) that have the potential to produce serious impacts but have not yet been discussed in detail in Japan. These themes included: climate security in the Asia-Pacific region, risks related to Japanese territory, social instability in Asia, risk perception in the business sector, risks to the supply chain and trade, and risks of food supply disruption. Our reading of these themes implies that future research should select the project funders (the MOE and the Environmental Restoration and Conservation Agency) as informants. These climate actors are very interested in understanding the structural underpinnings of climate security. Furthermore, influential domestically oriented ministries such as the Ministry of Land, Infrastructure and Transport and the Ministry of Agriculture, Forestry and Fisheries have also contributed to the securitization process in Japan.

At the time the project brief was published, previous climate security projects in Japan, unlike in the West, had not recognized the economic and socio-political risks of climate change as a compound risk with diverse consequences in national and international contexts, nor placed these impacts within a climate security framework or a broader security framework [10]. Climate security researchers in Japan need to catch up. In this connection, the multidimensionality of security, the interconnectedness of various aspects of security (and sources of insecurity), the embedding of (in)security in multiple institutions, and the costs of insecurity (to individual, community, or state) make security policymaking and action ethically sensitive, politically contested, and practically difficult across the world [18]. Mohamed Behnassi [19] alluded to this perspective in the sense that an examination of the utility of climate security as a framework for improving climate policy and governance revealed that when climate–security concerns are underestimated, new security challenges are to be expected in addition to existing challenges. Therefore, as climate security frameworks can enable security-sensitive responses to climate risks and create new dynamics in climate policy and governance [19], there has been growing attention to the challenges of compound extremes—multiple extremes occurring simultaneously [20]—entailing compound, interconnected, interacting, and cascading risks [21]. At the same time, given that the gap between the perceived and actual risk experienced by communities poses clear challenges for adaptation policy and practice, an awareness of climate-induced extreme events is not only a matter of community resilience [22], but also of societal resilience in the face of high-impact events, as it is particularly essential to understand the complementarity of these overlapping risks which aggregate elements of resilience, critical infrastructure protection, and climate adaptation [21]. The World Economic Forum’s *The Global Risks Report 2020* confirms the fact that “climate realities” represent an array of complex but unknown impacts [23]. Compound risk encapsulates these realities. Indeed, without taking into account complex interrelations with other policy areas, measures that are beneficial to the climate in principle could even have negative impacts [24]. Professor Simone Borg reiterated this complexity during an interview with Gaston Moonen. Climate change is not a simple environmental issue, but a complex one, and this is a challenge and potential prospect [25].

The purpose of the second research project (titled “Climate change and security: Filling remaining gaps”; https://bit.ly/3xx1MAa, accessed on 15 March 2022) was to both empirically address the idea of climate change/security and theoretically contribute to several themes relating to the associated discourse. Findings from this project were published in 2021 as a collection of articles in the journal of *Politics and Governance* (Table 1). Ten Japanese researchers contributed five articles to this collection, which were duplicated in the Google Scholar and ProQuest databases. Kameyama and Takamura [4], the editors of the collection, did their best to encourage contributions from all over the world (especially from China, Russia and India, but without success).

In parallel with this project-based collection, in which the issue of climate security policy is a prominent topic, the generalizability of our review may be constrained by the small sample size (34 in total), the focus on peer-reviewed literature, and the geographic focus on Japan. In light of the proliferation of systematic reviews over the past decade, many of which have focused on understanding issues and trends in specific regions [34], prior studies have warned that geographic constraints can lead to biased findings. In line with this warning, the review by Rocque et al. [13] focused on papers with a global focus and no clear geographic constraints. Because the results of survey samples focused on national or regional levels are necessarily constrained [12], readers may interpret this constraint as a weakness and thus point to the need for further research on climate change in general [35]. There is another important concern about geographic focus. Sietsma et al. [34] find not only topic biases by geographic location, especially regarding Western authors who are experts on climate issues in Global South countries, but also that criticisms of the Intergovernmental Panel on Climate Change’s (IPCC) tendency to emphasize natural sciences at the expense of social sciences likely reflect the rapid growth of social science topics and the preeminent position of natural sciences in adaptation research, not bias within the IPCC. These findings are revealed through the use of a novel, probing, computer-assisted evidence mapping method (*n* = 26) that combines structural topic modeling and expert interviews to formally assess the representativeness of IPCC citations [34].

There is a disadvantage to using geographic focus and peer-reviewed literature as key elements of the inclusion and exclusion criteria used to determine the literature of interest in a search strategy. We daresay that it takes an uncurious act of imagination not to imagine that this strategy may conceivably lead to an imbalance of studies (or *research imbalance*), which in itself may lead to study bias. Of importance here is a study by Klopfer et al. [15] which assessed the conceptual frameworks for climate impacts in urban areas through a scoping review of 50 publications and presented two key findings: the strong influence of IPCC publications and the imbalance in favor of European and North American researchers. This imbalance often invites criticism of research bias, but Klopfer et al. [15] do not discuss the issue of imbalance as a reason for research bias. To this end, the qualitative literature, due to its tendency to examine long-term environmental change, could be said to be biased (even when qualitative and quantitative methods are described in research papers), because this literature is often vague about the relative environmental origins of environmental degradation and the type (and/or magnitude) of change [12].

With regard to the peer-reviewed literature, previous research has provided general advice on the issue of bias and small sample sizes, warning against focusing on only peer-reviewed literature. There is an offset to be made here. Although limiting a systematic review to peer-reviewed literature has a high potential towards bias, because important trends and insights could be missed [11,13], a small sample size may be deemed appropriate because most systematic reviews of existing literature rarely exceed 100 documents [34]. In other words, we share a fact that can be found in the systematic analysis conducted by Vasileiou et al. [36]. In their review covering a more than 15-year period of qualitative studies, Vasileiou et al. [36] find that many review articles typically justify sample size by referring to practical considerations and the principle of saturation, which refers to the development of theoretical categories (as opposed to data) that become apparent when data collection no longer yields new insight.

We therefore believe that not all findings in a review based on a small sample size and single country focus can be bias-free and necessarily generalized to other world regions, although this type of review can offer important general theoretical and methodological insight for future research [12]. However, the focus on peer-reviewed climate security literature and a specific context (discourse and policy rather than policy instruments) from a specific country’s perspective allow for a narrower research focus. The national climate security context should be seen as complimenting international climate security research. While this complementarity offers an avenue for future research to advance what has been started here and fully answer the three research questions, Thurston et al. [37], noting their final selection of 37 articles from a global systematic review of 555 unduplicated records, acknowledge the need for more rigorous designs and better-quality studies to inform evidence-based policies.

Because avoiding avoidable bias is a central consideration in our review, we assessed the risk of bias in the included studies, paying particular attention to missing information. This consideration could only be productively accomplished through the examination of small sample size, as the number of hits in the preliminary database search was too large. Certainly, no matter what methodology or analytical method is employed, comprehensive review studies cannot be completely free of bias. Future studies may follow our example and strive to considerably reduce the possibility of inevitable bias. Researchers can reduce the likelihood of presenting biased information by verifying important details that are vaguely conveyed or omitted entirely in the publications they choose to study. For example, we had difficulty identifying the method used by Ide et al. [31] in their study, so we contacted them and found, unsurprisingly, that it was a mixture of case study, theory building and literature review. As another example, a systematic review by von Uexkull and Buhaug [38] (p. 10) states: “Schmidt et al. (2021) shed new light on the understudied interstate dimension of climate security. They show that anomalous climate conditions can shape the risks for new diplomatic conflicts and militarization of ongoing issues, and that chances for issue claims and conflict initiation are greatest for revisionist states.” Reading the paper by Schmidt et al. [39], one is unfortunately surprised to find no mention of climate security, climate and security, climate/security, or climate–security.

To end this section, it should be noted that while the literature corpus on climate security is now too vast to examine in a manageable way, an impact assessment of the expanding and already large literature on other intertwined concepts or closely related terminologies is rapidly becoming unrealistic. Of course, with increased sample size being a better indicator of general trend, the search of all relevant keywords in abstracts would likely yield more papers relevant to the focus of investigation, suggesting the search will lead to an unmanageable amount of literature to review. Still, the suitability of a systematic review process cannot be ignored because it can lead to greater transparency, reliable replicability, more insightful conclusions, and better comprehension [15], especially if sufficient care is taken to minimize bias.

### 3.1. Has Japan Participated in the Global Climate Security Debate at All?

Although Japan “needs to start taking part in the climate security discussion” [10] (p. 20), Tokyo has participated in and contributed to the climate security debate through research and policy discussions because countries around the world are expected to do so. Future researchers need to make a neat distinction between Japan’s participation and contributions. To begin with, contribution (which may be minor or substantial) means adding value to the debate and participation means taking part in the debate (whether actively or passively); researchers should set boundaries for what is and what is not participation, and divide participation itself into relevant categories. Participation in this sense implies active involvement in decision-making at all levels and concerns the formulation and implementation of climate policies. Thus, participation implicitly implies that climate security research will have a substantial, if not permanent or explicit, impact on policy debates. In other words, given the current reality in Japan, there is little room for leveraging influence when climate security research is scarce.

Therefore, given Japan’s track record in green innovation and resolute determination to coordinate and lead the innovation of technological approaches to reduce global warming, one may well expect further breakthroughs in the near future. According to Naoko Ishii, Director of the University of Tokyo’s Global Commons Center and the government’s Climate Policy Adviser, the greatest challenge facing humanity is to balance the natural system of the Earth, of which we are a part, with the interdependent economic system that exists on top of it [40]. What can be achieved in this context is the likes of the Fukushima Hydrogen Energy Research Field (a cutting-edge innovation developed through a collaboration between five private companies and the New Energy and Industrial Technology Development Organization), which is the world’s largest hydrogen plant using renewable energy. Japan sees such solutions as central to winning the fight against climate change.

This is a testimony to Japan’s unwavering commitment to continue to participate and contribute to the global issues covered by environmental security and climate security. In 2007, in a report recommending policy benefits of climate security at home and abroad, the MOE highlighted that Japan’s position on the concept was due to research into four notable developments: the security concept, the comprehensive security concept, accelerating climate change, and the international debate on climate security [24,41,42]. It was also in 2007 that the effects of climate change were first discussed in the UN Security Council. It was in 2008 that the European Commission submitted its report on climate change to the European Parliament. Furthermore, it was not until 2010 that the Obama administration mentioned the threat to national security posed by climate change in its Quadrennial Defense Review (QDR). The 2014 QDR published by the United States Department of Defense acknowledges climate change as an escalating threat, suggesting that it may complicate a series of threats to security and stability. In other words, the Japanese government’s initiation of discussions on climate security was not a late move compared to international trends.

The way global society conducts its daily life is a major contributor to environmental problems and one of the main reasons why environmental security is needed. Particularly in Japan, where civil society organizations are generally led by male leaders, the number of organizations representing women’s interests is limited to about 2.2% [43], and civil society does not have full and unrestricted access to policymaking processes; thus, a comprehensive policy on environmental security monitored by civil society could have a considerable impact. There is a fundamental belief about the latter perspective. That is, according to Loada and Moderan [44], civil society participation is central to three key functions. First, it strengthens the governance chain of security. Second, it provides democratic political requirements that strengthen the legitimacy of decisions affecting individual security. Third, it provides strategic requirements to strengthen the necessary national ownership of the process and thus the transparent governance of the security sector.

The concept of environmental security, little known in the immediate post-Cold War period, has been adopted in Japan as part of human security and remains at the core of Japanese diplomacy [28,45]. When the UN Development Programme put forward human security in its 1994 Human Development Report, the Japanese government responded favorably. Keizo Obuchi, in particular, strongly supported the idea of human security. In his speech in Singapore in May 1998, when he was Minister of Foreign Affairs, he officially announced for the first time that Japan would incorporate this idea into its foreign policy. He also emphasized the importance of human security in two policy speeches during his tenure as prime minister from 1999 to 2000. Since the late 1990s, the Japanese government has actively lobbied the international community for the realization of human security. When the UN Secretary General Kofi Annan visited Japan in 2001, Japanese Prime Minister Mori proposed the creation of a “Commission on Human Security” co-chaired by Sadako Ogata, the UN High Commissioner for Refugees, and Amartya Sen, the President of Trinity College, University of Cambridge, and Nobel Prize winner in economics. After two years of discussions, the committee worked to develop a concept and guidelines for human security, and submitted its report to Prime Minister Koizumi in February 2003 and to UN Secretary General Kofi Annan in May of the same year. Subsequently, Japan has worked to disseminate the concept through such efforts as the launch of the Friends of Human Security meeting in 2006 and contributing to the adoption of the first UN General Assembly resolution on human security in 2010. In addition, the Japanese government contributed approximately JPY 500 million to establish the Fund for Human Security at the UN in March 1999. Since then, the Japanese government has contributed a cumulative total of approximately JPY 47.8 billion to the Fund through FY 2019, supporting a total of 257 projects in 99 countries and regions [46].

Therefore, it is inaccurate to say that Japan has not meaningfully considered nor discussed climate change as a security issue. In fact, the latest research findings on climate security in Japan attest to this fact. Specifically, in his paper titled “Transforming the Dynamics of Climate Politics in Japan: Business’ Response to Securitization”, Takahiro Yamada [26] reiterates the need to revisit the question of when the implementation, and thus the securitization, of climate security is considered successful. In summary, future researchers should note that other contexts may also need to be revisited, fine-tuned and repackaged. In this context, existing research on the pivotal position of Keidanren—the powerful Japanese confederation of industries—in Japan’s climate policy needs to be upgraded. Researchers may want to showcase why Keidanren is not merely an influential actor but a policymaker to reckon with. Its website is a mine of information waiting to be exploited, especially in relation to the decarbonization agenda (see [47]).

### 3.2. Why Did Climate Security Struggle to Become a Major Political Theme in Japan until 2020?

For reasons of space limitations, it is impossible to cover every major factor that may well explain this question and thus the slow progress of climate security research in Japan. To begin with, the role of certain ministries is particularly noteworthy in Japan, where the Ministry of Economy, Trade and Industry (METI) and the MOE are the heavyweights in climate change policymaking. The Ministry of Foreign Affairs (MOFA) oversees other ministries when they contribute to climate change policymaking, pertains to Japan’s position at the international level, and helps build inter-ministerial consensuses. The Ministry of Land, Infrastructure, Transport and Tourism (MLIT) will always play a role in any discussion involving emissions from the transportation sector.

Few (if any) previous studies have examined perhaps the four most important among these factors (the scarcity of systematic research on climate security in Japan, regional imbalance, the METI–MOE inter-ministerial rivalry, and thematic imbalance or the dynamism between environmental security and climate security). In relation to the scarcity of research, Kameyama and Ono [1] and Kameyama and Takamura [4] provide insight into this issue. To shed light on this, Kameyama and Ono [1], arguing that climate security seems to have become more popular in Japan since around 2020, trace climate security discourses to determine why so little exists in Japan before this period and whether such discourses could inform more comprehensive responses to the climate change issue. According to Kameyama and Ono, two approaches to climate security (short-term sudden risks to individuals and long-term irreversible global change) have been considered in Japan, even though the term climate security is not specifically mentioned, but the other two approaches (impacts to military and defense organizations, and causes of conflict and violence) have not been discussed and need to be incorporated in the discussions in Japan. They also clarified that these omissions are partly due to the different interpretations and usages of the term climate security in the existing literature.

Future research can make an important contribution by considering the following four factors, as there is a scarcity of research works that do so in a single paper. First, the scarcity of systematic research on climate security has affected policy progress in Japan. In the collection edited by Kameyama and Takamura, four contributors urge future research to provide more “systematic” examinations of and explanations on climate security. We belief, as Kameyama and Takamura [4] write, that no matter how climate security is conceptualized theoretically, the concept ought to be fully deployed as a tool to promote not simply coordinated collaboration but, as Ide et al. [31] point out, the gathering of more systematic data in order to better clarify the referent in need of protection and thus whose security is at stake. The same can be said for Räisänen et al. [29], who argue that climate risks are not always followed in a systematic and coordinated way, and for Jakobsson [33], (p. 16) who not only “systematically” answers her research question but takes issue with existing research on the ground that—despite scholarly focus on normative dimensions, institutional expansion, securitization and the conceptualization of discourse—not a single study covering the 2007–2010 period attempts to systematically articulate why the policy debate on climate-related migration surges in this specific period. In parallel, bearing in mind that some degree of complementarity always exists between academic contribution and policy initiative, Hardt’s [32] pro-logic theory in the conclusion section of her paper states that future research needs to investigate the scientific research on the climate–security nexus more systematically because the main articulations on this nexus ignore key scholarly articulations of climate security, especially the Anthropocene dimension that is yet to become a topical topic among Security Council members.

The second factor is what Kameyama and Takamura [4] identify as the problem of regional imbalance. This problem stems from the fact that most of the literature on climate change and security published to date has been written by authors from North America and the European Union [4,15,34]. Kameyama and Takamura urge for more voices from other regions (such as Asia) because there are few experts and few case reports from these regions. This is an obvious lack of knowledge that deserves attention, and the scarcity of case reports that could compensate for this lacuna make research such as the present study timely and compelling. Although this claim only reemphasizes a prominently highlighted lacuna in existing literature, on the other hand, we contribute by suggesting the issue of regional imbalance as a major cause of the paucity of climate security discussions in Japan. Evidence of this suggestion can be supported with related statements made by some contributors to the collection of articles edited by Kameyama and Takamura [4]. For example, bearing in mind that the tendency of cases focusing on particularly vulnerable regions or large countries is suggestive of the need for long overdue wider geographic coverage [29], climate security has been accorded low priority in Japan compared to the academically trending status of this concept in the West [4,26], and “the recent emergence of security-related climate change rhetoric in Japan has not yet been covered in the academic literature” [27] (p. 54). Hasui and Komatsu [28] are quite explicit: the perspective of climate security is rare in Japan. There are two avenues at scholars’ and policymakers’ disposal when seeking to further address the problem of regional imbalance. First, commissioning more systematic, country- or region-focused projects, and second, increasing the participation of non-European researchers in such projects (this study is a clear example of how to meet both requirements).

We suggest there is high likelihood that progress on climate security has been (and is being) impeded by the METI–MOE inter-ministerial rivalry (the third factor). Previous studies referring to this rivalry have attempted to answer the question of why climate security has struggled to become a major political theme in Japan. In recent studies, Koppenborg and Hanssen [27] only briefly touched on the role of political parties, while Hasui and Komatsu [28] addressed hypotheses such as the lack of dynamism among parties and the dearth of scholarly interest in climate security. With more disagreement than agreement so far, not only is there room for improvement in these areas if future research is to satisfactorily answer the discussed question, but it is also necessary to correctly understand the rivalry which, to some degree, may not be separated from domestic climate politics and is actually related to the involvement of political parties in the policy process.

Despite this rivalry and the METI’s and Liberal Democratic Party’s consistent support for industry’s lack of enthusiasm toward decarbonization—which has dictated the tone of Japan’s basic energy plans [26]—policy action on climate security should progress. This perspective is congruent with Hasui and Komatsu’s [28] description regarding the introduction of a unique climate security concept into security policy in Japan as an enabler of preserving environmental protection and national security values. While that unique development represents, at a minimum, an obvious securitization move, climate securitization is a difficult process partly because the success of a securitizing move depends on a variety of factors. According to Yamada [26], one such factor is the mismatch between the scale of the climate crisis and the identity of the actors tasked with addressing this crisis, as the Japanese case seems to exemplify rather explicitly. This is another interesting insight to note, partly because in the same collection in which Yamada makes this claim, Koppenborg and Hanssen [27] identify civil society, city, parliament, and government as key actors that frame climate change in Japanese climate policy. They cite several framings made by these actors. For example, the Japanese Diet states (in its declaration on climate emergency): “we share the global perception that ‘the global warming problem now has exceeded the realm of climate change and entered into a situation of climate crisis’” [27] (p. 60). Future research needs to explain how the operationalization of the concepts of power and securitization drives who can and who cannot participate in the securitization process. It is equally important to explain what subalterns need to do to gain access to the privileged securitization arena where their fate is often debated and rubber-stamped in relation to the climate measures needed to protect them. It should also be noted that a more practical way to make climate action more effective is to define climate security in a universal way.

As Japan embarks on its decarbonization journey, there has been a growing increase in the number of authors discussing the METI–MOE rivalry. However, it remains unclear whether the METI or the MOE is the most influential actor in domestic securitization. The MOE, as Yamada [26] recounts, is considered a prime candidate for a domestic securitizing actor because the MOE-led coalition of securitization-inclined government officials has created a pre-emptive regulatory environment for decarbonization. This is a clear position on climate security. As this coalition has adopted a securitization formula that calls for special measures outside of normal political procedures, establishing climate securitization as a new minor discourse [27], the MOE has implemented policies aimed at social change for disaster reduction [28]. Policy implementation may have benefitted from a *special privilege*.

Based on the papers published in the collection edited by Kameyama and Takamura ([4], see especially [26,27,28]), studies on climate securitization typically present the MOE, the METI—often in favour of Keidanren’s position on climate policy and vice versa—and the MOFA as key actors with the authority to address climate change and develop national climate policies, including climate security policy. With the government largely taking a back-seat role where the METI advocates energy efficiency and economic growth, and as the MOE pushes for strengthened environmental climate action, the power tussle often favors the METI and the (heavy) industries it represents [27,48,49]. Even so, this is not always the case. The prime minister and the foreign ministry sometimes supported MOE’s climate policy decisions over those proposed by the METI. For example, in 2008, the MOE wanted to set the emission reduction target at the national level, but the METI strongly opposed the idea, resulting in then Prime Minister Fukuda’s intervention which decided that Japan would accept a certain level of target [48]. Still, while the administration of the then Prime Minister Shinzo Abe and Chief Cabinet Secretary Yoshihide Suga usually supported the METI [49], other important actors included the Ministry of Defense (who have not yet substantially addressed climate change countermeasures [28] but established an internal task force on climate change in May 2021), the Ministry of Agriculture, Forestry and Fisheries, and the Ministry of Land, Infrastructure and Transport.

Given that these influential ministries do not mutually define climate security, researchers need to be more explicit about the timing, rationale and procedures of the major climate actions implemented (or likely to be implemented) by each ministry, taking into account the distinctive features of the climate security debate. Although the MOE’s articulation could be seen as the beginning of a mutual definition, more research is needed on three important fronts. First, outlining how the lack of a widely accepted definition has hindered clear policymaking on climate security in Japan. This difficulty arises not only because climate security and environmental security require more comprehensive and systematic research; the mainstream approaches adopted by Japanese researchers diverge from the main approaches to environmental security policy research in the West [28,42]. Second, ensuring that climate measures contribute to climate security for all involved countries, to initiatives in line with Japan’s notion of climate security, and to protecting citizens from climate threats [41]. Third, identifying insights that may be gleaned from Japan’s emphasis on a bottom-up approach to policymaking, because it seems to be the case that policy feasibility seems to be more important than ambition [49].

Ken Sofer [50] provides a compelling empirical account of climate politics and the development of Japan’s energy goals, and how these forces both complicate and provide opportunities for cooperation on the climate crisis. Based on an analysis of interview transcripts, he examined the role of political parties, parliament, public opinion, the decision-making process of the bureaucracy, and the balance of power between the interest groups. In what Incerti and Lipscyv [51] label Abenergynomics, the Abe Shinzō administration adopted energy policy as a tool to foster the economic growth goals of Abenomics, even when the associated policies were clearly unpopular with the public, opposed by the power companies, or very toxic to the environment. During that period, the METI–MOE rivalry over climate policy intensified. Especially since the 2012 elections, these rivals, and to a lesser extent the MOFA, have had the privilege of almost complete control and management of Japan’s domestic policy (environmental and energy policy) and international climate policy, with minimum input from both the party leadership and the Diet or the Prime Minister’s Office (known as the arbiter that keeps rivals from each other’s throats). In this “unhealthy” competitive environment, Japan’s leadership position in global climate governance has been severely diminished and this has been detrimental to the status and progress of domestic climate security action. Interestingly, this has contributed to Tokyo’s desire to regain its leading position in global climate governance. Charity, as you will agree, begins in one’s own country or regional affiliation. With this in mind, Japan can regain the lost position by setting up more climate initiatives and following them up decisively.

As a better alternative to previous studies, future studies should complement Sofer’s useful insights. In particular, the focus on climate security [27,28] and climate politics/policymaking [49,50,51,52] allowed these authors to provide more detailed information. However, combining these topics into a single manuscript would help the reader to infer their relevance in the context of the issue of the universal definition of climate security, which may not be completely separable from environmental security. Climate security and environmental security have been extensively studied by Western researchers. They have been understudied in Japan. Prudence is important when exploring very early studies on environmental security and climate security that were published in Japanese [53,54,55,56]. This is because these two concepts do not always lend themselves to simple understanding.

The fourth factor is the dynamism between climate security, environmental security and human security. Given that these concepts are essentially true to the spirit of protecting human needs, it is possible to reasonably make similar comparisons between them while acknowledging, in terms of the intersections between the concepts, content and structural differences between them. This dynamism is useful, but it can sometimes display what we call thematic imbalance. The idea that climate change will adversely affect the material and social stability necessary for preserving human security is part of the concept of climate security, which is generally included as part of the human security agenda in Japan [9,10]. The tendency to consider climate security as part of the latter, and sometimes as part of environmental security, has not helped in clarifying what may be perceived as confusion whenever these different concepts are interchangeably implemented in Japan. This *policy practice* is common when these concepts are not properly distinguished, this is precisely what is meant by thematic imbalance. The first research project co-funded by the MOE and the Environmental Restoration and Conservation Agency contributed to this point by reorganizing those concepts, including important related ideas, and applying them to Japan and the Asia-Pacific region. By moving beyond dominant narratives on concepts to the emerging decarbonization agenda, future research should rigorously and systematically discuss the level and extent of Japan’s participation in the climate security debate, including contributions (if any) to defining climate security. It is also essential to fully explain the possible reasons why climate security is less familiar in Japan than in the international community. It is hoped that future qualitative researchers will fully elucidate thematic imbalance, including what needs to be done to avoid it. One way to do this is through thematic analysis, which is useful for interpreting trends in textual data (especially interview transcripts).

On this point, defining climate security and environmental security has long been of interest to some Japanese researchers. A good example of closely related discussion can be found in research published by Norichika Kanie [56] in 2007 (whose title translates into “Towards the formation of climate security: The truth of high-politicising environmental politics”). Kanie suggests the importance of “structural climate security” to explain the need for a new world order and new international institutions based on it. While there is a move to prioritize frameworks for cooperation between countries in the new world order, Japanese policymakers begun discussing climate security at the International Climate Change Strategy Global Environment Committee meeting (within the Central Environment Council) in February 2007 [42]. This is a very interesting testimony to the fact that Japan, as a global actor, has sought to meaningfully address climate security.

Another noteworthy example is Seiichiro Hasui, who published his paper in English. Hasui [42], highlighting that the same argument can be made when considering the different concepts of climate security and environmental security, noted that the discussion of climate security on the global stage began with the remarks of the then United Kingdom Foreign Secretary Margaret Beckett at the UN General Assembly in September 2006 and at the UN Security Council in April 2007. Hasui believed that climate change security theory is very complex and that its meaning can change slightly depending on who uses it, and classified such meanings into the security *of climate, through climate policy, for climate protection and problems caused by climate change.* While these definitions were made in 2011 and remain largely the same and valid today, the latter—security problems arising from climate change—should be prioritized in future research, and not merely because of their socio-political value and disabling effect on the economy. This is particularly so because climate-related environmental change and degradation remain a major concern. All this should receive more attention in future research.

### 3.3. Why Did Japan Explicitly Start Dealing with Climate Security as a Policy Issue in 2020?

There are several answers to this question. A series of major events in the global arena likely caused a sudden heightened interest in climate risk in the 2020–2021 period in both the Japanese public and private sectors. A remarkable shift in the global climate debate with regard to the development of new dimensions of public discourse in 2019 likely produced a similar effect. Across the world, a wave of environmental and climate emergency declarations simultaneously emerged around this period [24,27], paralleled by another notable development: the United Kingdom became the first country to legislate a zero-emission target for 2050 [28]. Researchers have noted similar developments in the past. For example, according to a policy piece by Anthony Giddens [57] in 2008, climate issues have remarkably and rapidly moved to the center stage of public debate, generating vast literature from scientists, other scholars, and journalists. Giddens also notes that, as scientists continue to express serious concerns about global warming, the issue now receives almost daily media coverage; however, environmental groups are struggling to push governments and citizens to take the issue more seriously.

Other reasons that may well explain this development include, first, the emergence of a global decarbonization agenda, and second, when it comes to a decarbonized society, there is a growing realization that climate change has entered the climate crisis phase in all areas of Japan, and that the METI and the MOE are, in the words of Shinjirō Koizumi, “not enemies’’ [27] (p. 60). Multiple dovetailing factors contributed to that development. For example, the international debate on climate and security (both policy and academic), the UN Security Council’s interest in climate-related security risks, national declarations on climate security, the nascent (but budding) political interest in climate security in Japan, the influence of climate securitization on the position of the Japanese business community, Japan’s determination to maintain a leading role in further promoting and implementing its human security agenda, and the link between nature-based solutions and climate security. There has also been growing momentum in the private sector for companies to disclose climate risks as part of international financial reporting standards. In this context, Kameyama et al. [58] contributed to the first research project led by Kameyama [10]. They interviewed 11 Japanese companies to ascertain their perceptions of climate-related risks. While it is difficult to accurately assess the exact impacts of climate change and the extent of such impacts, the results showed that companies operate in an era in which investors are expected to prepare for these risks, have a high awareness of some risks, and are preparing or considering ways to confront them; however, some companies do not consider climate risks [58]. That said, the Paris Agreement became operational in 2020, and President Joe Biden made climate change a central issue in United States foreign policy in 2021.

In today’s academic and policy spheres, it is becoming increasingly popular to discuss climate security in terms of global decarbonization and energy transition. However, diverse conceptualizations of decarbonization, low carbon transition, low carbon development, and the interrelationships among these concepts have created unclarity [59]. Moreover, there could be security implications of responses to climate change in the form of low carbon development [60]. The scoping review by Klopfer et al. [15] conveys an important message: ongoing shifts imply the need for a comprehensive overview of how climate change impacts have been evaluated. This implies that only a limited number of (non)systematic reviews have been conducted on conceptual frameworks applied to evaluate these impacts in urban areas, and therefore implies the difficulty of obtaining the main directions and key methods in closely related areas. In parallel, Pham and Saner [11] point out that the notion of inclusiveness has been underrepresented with respect to both sentient nonhuman animals and the interests of future generations, and emphasize the need for conceptual and empirical research because of the lack of reviews focused on the issue of inclusive climate adaptation.

Japan’s sudden legalization of its decarbonized target in May 2021 was in part a response to notable developments at the global level. Nonetheless, focusing on the Indo-Pacific region (where Japan remains a major geopolitical player), some innovative research projects and policy initiatives led by Japan should enhance our understanding of the decarbonization program. For example, Japanese Prime Minister Shinzo Abe launched the “Free and Open Indo-Pacific” foreign policy strategy in 2016, presenting it as a key for the prosperity, peace and stability of the international community through the dynamism of integrating two continents (fast-growing Asia and Africa with its enormous growth potential) and two oceans (the Pacific and Indian Oceans). With this strategy, Japan seeks to expand its diplomatic clout by promoting the Indo-Pacific region as a free and open international public good, as an active participant in the rules-based international order, and, in particular, as a region that will benefit from Japan’s readiness to foster greater cooperation and coordination in the area of disaster relief and humanitarian aid provision.

Due to the Indo-Pacific region’s increasing challenges, such as human security and climate issues, Japan’s Indo-Pacific strategy is all the more important. This strategy is intensifying at a time when the United States and its allies are quickly shifting their gaze to the region, which: covers the Indian Ocean and the Pacific coastline; is the typical residential site of more than half of the global population; is home to almost 65 percent of the global economy; and is the epicenter of the climate crisis and thus central to climate solutions. However, Japan is not the sole innovator of the Indo-Pacific strategy because the notion of an Indo Pacific-based economic corridor was mooted when India and the United States held a strategic dialogue in 2013. On few occasions thereafter, the then Prime Minister Shinzo Abe talked with Indian Prime Minister Modi about how to progress this notion. In February 2022, the long-expected United States’ Indo*-*Pacific strategy was announced by the Biden administration. The United States’ strategy, similar to Japan’s strategy, comprises an economic framework that will assist countries to adapt to the climate and energy transition, among others. According to the White House [61], Indo-Pacific countries are key players in defining the nature of the international order in today’s rapidly changing strategic landscape, and the United States and its partners worldwide have a stake in the outcome.

Useful in this context is a project sponsored by the Sasakawa Peace Foundation, Japan’s largest private foundation. Nine scholars contributed to the main outcome of this project: a book entitled *Climate Security: Global Warming and A Free and Open Indo-Pacific* that was published in 2021 [62]. In addition to this book being the first published on this subject in Japan, Yasuko Kameyama, Director of the Center for Socio-Environmental Systems Research at the National Institute for Environmental Studies, and Keiji Ono, Research Fellow at the National Institute for Defense Studies, published a co-authored paper on climate security in Japan in an international journal [1] and issued a press release in October 2020 titled “What is climate security?” In addition to these scholarly contributions being one of the reasons why the Japanese society gradually became interested in this topic, the Nikkei and Asahi, the two influential Japanese newspapers, published a series of column articles on climate security one after another in April 2021. In May 2021, the Ministry of Defense newly established an internal “Climate Change Task Force” under the Senior Vice Minister to study the impact of climate change on security. It can be said that 2021 was the starting year for climate security discourse in Japan. This is similar to the establishment of the “Climate Diplomacy Task Force” within the MOFA in May 2018 to strengthen cross-sectoral efforts within the Ministry and promote climate change diplomacy more actively and effectively [63].

Additionally, of relevance is Tangney et al.’s [64] explanation regarding why the Indo-Pacific region, due to ongoing socio-political and environmental challenges as well as existing tensions within and among neighboring countries, is an important case study for understanding two important considerations in terms of climate security. According to the systematic review and the synthesis of academic and grey literatures conducted by Tangney et al. [64], these considerations are, first, the interdependence of climate adaptation and regional security challenges in the Indo-Pacific region, and second, the four main themes that are prominent in discussions and analyses by governments, NGOs, and academics. These themes consist of the maintenance and enhancement of climate security in the Indo-Pacific, governments’ abilities to identify and resolve climate maladaptive pathway dependencies, governments’ abilities to manage finite resources efficiently, and governments’ readiness to cooperate despite inherent tensions [64]. In parallel, Sharifi et al. [65] identify four major thematic focus areas, namely, climatic impacts that may lead to large-scale human displacements, war and violent conflict, institutional mechanisms for addressing political tensions and conflicts, and cooperation/conflicts related to water resources.

This is an important reason why Japan should continue prioritizing a clear, focused, and decisive involvement in the global climate security debate as an indispensable aspect of its domestic climate change agenda (see Table 1). Especially in some Indo-Pacific countries that are characterized by unstable governance structures and questionable governmental legitimacy, a variety of challenges may negatively affect regional security in the form of cascading climate security concerns, such as ongoing human security issues related to resource conflicts and forced migration between jurisdictions [66,67]. Against this background, the climate geopolitics of the Indo-Pacific region is directly relevant to Japan’s climate policy and will become more integral after 2022. With the Indo-Pacific being a region where climate-induced conflict within and between countries can affect the stability of communities and institutions in the greater regions, studying this region’s climate politics is an effective way of bringing more attention to both the political dynamics in the region and the larger framework of global climate governance that the region offers [68,69]. This perspective is particularly useful, partly because ensuring human security is the cornerstone of the Free and Open Indo-Pacific diplomatic strategy. In many cases, opportunities remain for responsiveness to decision-making involving the environment and security interactions more holistically, even if future scenarios on climate politics, socio-economics, and their impact on climate security links are highly uncertain [17,70]. Important in this regard is the systematic review of 1,337 articles conducted by Sharifi et al. [65]. One of their major findings emphasizes how three decades of research on the interactions between climate change and negative peace-impairing events has focused primarily on war and violent conflict, with little focus on other important developments. Conceivably, while one of these developments is Japan’s Free and Open Indo-Pacific diplomatic strategy, this finding not only informs climate security research and policymaking but is also consistent with Pearson and Newman’s paper. Pearson and Newman’s [17] (p. 2) systematic review contributes to “the broader and expanding field of climate security” by pinpointing the difficulty of neatly categorizing types of conflicts and the various groups—sometimes unclearly documented—identified by previous literature. Perhaps more interesting is Williams and McDuie-Ra’s [68] (p. 10) argument: there are different narratives of climate security which are “also constructed as vulnerabilities that undermine human security”.

Table 1 is compelling for future research, as it derives important findings for advancing the global climate security debate and Japan’s involvement in it. With this table in view, future studies following Williams and McDuie-Ra’s [68] argument would be compelling not only in relation to the table but especially for fresh information on how climate change has been constructed and politically shaped by national, regional, and global climate politics, which in turn has shaped the climate change debate.

## 4. Discussion

At a time that Security Council member and non-member countries continue to debate the security implications of climate change which is being altered by various conceptions of the climate security framework, which in turn is being altered by how actors perceive the urgence of addressing climate change, policymaking communities are increasingly acknowledging the Security Council’s relevance in resolving climate security challenges. At the same time, unpacking the debate remains a major preoccupation for actors in the academic and policy fields. For example, research on global climate security governance has not stopped at discovering the importance of research issues such as those raised in this review paper, but researchers are yet to widely acknowledge this particular research as a field in its own right. For context, we acknowledge the lack of a normative and universal definition of climate security, which has only recently become a sudden trend among Japanese academics and political elites. The lack of a widely acknowledged definition seems to be one of the main reasons why climate security is not yet a major political issue in Japan. On the other hand, as suggested in the previous pages, the answers to the three research questions imply that the climate security framework is robust at the international level. We hope that similar robustness will be ensured in Japan in the near future.

Indeed, our review of very recent studies on the current state of climate security policy in Japan has revealed that future researchers should be able to complement our answers to three key questions that will determine the direction of future climate security research. Despite a preliminary analysis of data from the climate security literature, these questions remain unresolved and are largely unknown nationally, which is why we propose further study on this gap. A small group of Japanese researchers have briefly touched on hypotheses concerning the lack of scholarly interest in climate security, the very nascent MOE-led securitization discourse, and the lack of dynamism between political parties. However, also evident is the need to further review academic papers as well as domestic and international policy documents, because simply reviewing the previous literature can never be sufficiently informative for policymaking purposes. Interpreting empirical data will make future contributions more impactful. The best source of empirical information is to interview political elites and the ministries responsible for responding to the climate crisis. Sitting down with Japanese government officials (especially from the Ministry of Land, Infrastructure and Transport, the Ministry of Defense, the Ministry of Agriculture, Forestry and Fisheries, the MOE, and the METI) and hearing them talk about their climate security-related concerns and plans, including their views about conceptualizing a universal definition of climate security, should lead to much needed insight. While gaining such insight requires asking for information about the emerging pathways to decarbonization, we envisage that some researchers may not have access to these sources for whatever reason, such as time or conditions of access. To compensate for this constraint, it may be necessary to explicitly justify the relevance of the case study method in climate security research.

## 5. Conclusions

As enumerated the in previous pages, readers should be reminded that the overarching purpose of our review paper was to extract some implications for a deeper examination of climate security in Japan, rather than providing a complete solution to the three questions mentioned above. Overall, Japanese researchers’ main arguments are not sufficiently persuasive or empirical, partly because they have not attempted to examine these three questions in more depth. Future studies will no doubt extend existing research by exploring key definitions of climate security and climate securitization, especially if qualitative researchers unpack these definitions in a single manuscript. Researchers will be able to provide important and more robust insight by assessing the extent to which Japan has securitized the global climate security debate, why climate security struggled to become a major political theme in Japan until 2020, and why climate security became a clear policy issue in Japan in 2020. To be certain, future research will make important contributions to knowledge by examining the following topics: the extent to which Japan has securitized the global climate security debate; why climate securitization struggled to become a major political theme in Japan until 2020; and why Japan began to explicitly address climate security as a policy issue in 2020.

## Figures and Tables

**Figure 1 ijerph-19-08253-f001:**
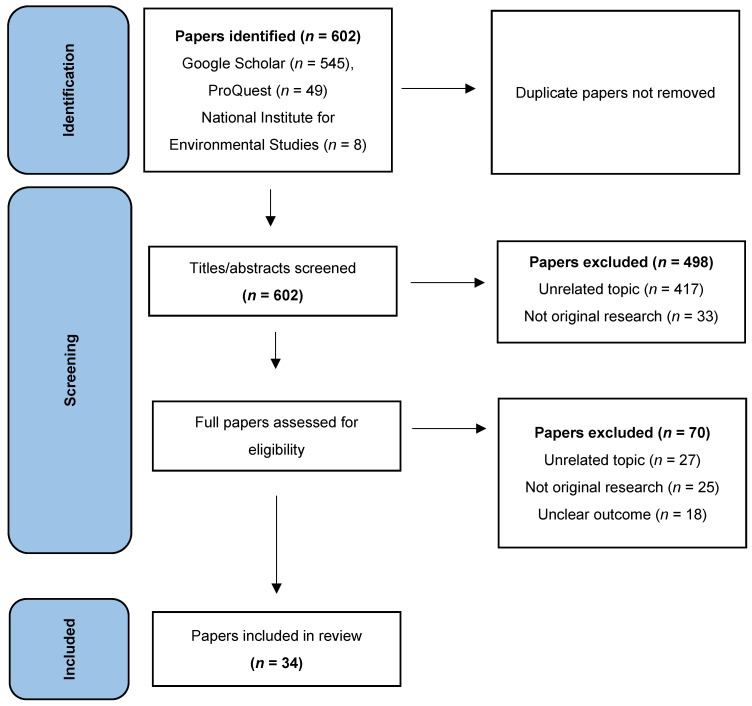
Identification of papers from databases.

**Table 1 ijerph-19-08253-t001:** Key findings from the second research project.

Contributor	Level/Focus	Method	Key Finding
Kameyama and Takamura [4]	Global/Climate change and security	Introduction to the collection	Climate change and security lack a single agreed definition
Morita and Matsumoto [5]	Asian region/Nature-based solutions governance in Asia	Case study	Nature-based solutions tie into climate security discourse
Yamada [26]	National/Climate securitization in Japan	Text-mining	Collective securitization by the Ministry of the Environment-led coalition
Koppenborg and Hanssen [27]	National/Situating Japan in the international climate security debate	Discourse analysis	A burgeoning securitization discourse
Hasui and Komatsu [28]	National/Contextualization of climate security within Japanese climate and security policy	Literature review	Key characteristics of climate security considerably overlap with the transition of Japan’s security policy
Räisänen et al. [29]	National/Finland’s comprehensive security policy	Content analysis	The Finnish comprehensive security model is a holistic perspective on security and can take environmental security concerns into account
Prabhakar et al. [30]	Asia/Climate Fragility Risk Index (CFRI) and critical threshold concept	Development of CFRI	Climate security and CFRI can complement external emergency assistance
Ide et al. [31]	Global South/Gender perspective and the climate-conflict nexus	Mix of case study, theory building and literature review	Gender concerns should be central to future climate conflict research
Hardt [32]	Global/United Nations Security Council and climate security	Case study	An incoherent consensus on the passively shared mainstream conception of climate–security nexus is being established
Jakobsson [33]	Global/Climate-induced migration	Case study	Climate-induced migration is related to the climate security debate

## Data Availability

No new data were created or analyzed in this study. Data sharing is not applicable to this article.

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
