# Peer review of "A Review of Climate Security Discussions in Japan"

_ijerph, 2022, doi:10.3390/ijerph19148253_

Round 1
Reviewer 1 Report
Many thanks for the opportunity to read this article. The systematic review presented in the article yields some interesting insights to climate security research overall and particularly to the climate security discussion in Japan. However, the methods and especially the materials used do not seem to fully correspond to the research questions that the paper aims to solve. As a result, it is not quite clear what is the original research contribution of this paper. In addition, there appear to be some relatively significant issues with regard to the definition of very central concepts, starting with climate security itself. Therefore, I would unfortunately suggest rejecting the paper.
More specific comments follow below.
First, the linkage between the Japanese policy discussion on climate security and the academic research used as the material of the paper remains unclear. The authors present their research questions as: 1) Has Japan participated in the global climate security debate at all? 2) Why did climate security struggle to become a major political theme in Japan until 2020? 3) Why did Japan explicitly start dealing with climate security as a policy issue since 2020?
These all appear to be questions that should be addressed through materials such as policy documents, interviews or even newspaper data. The authors do very comprehensively justify systematic literature review as a method, and even the case for a small sample-size review. However, in this case, due to the small sample size specific research questions, the actual analysis in many cases only ends up summarising results from individual previous research papers. In addition, the sample seems to include several papers that do not analyse the Japanese case at all, but instead focus on some other geographical case or are global in scope. While these would be useful as previous literature, aiding the analysis, it seems inappropriate to include them in the sample addressing the research questions of this paper.
Overall, the questions appear to be addressed on the basis of a very small number of studies that are summarised, often replicating the questions that these papers have initially studied. The authors do propose further research needs on the basis of this analysis, but this is a very limited research result. Therefore, the unique research contribution of the paper remains weak.
The authors do also justify the use of literature review by pointing out that research is a precondition to policy-making. This is a fair argument, but the authors seem to assume a very direct and strong relationship between the two. They, for example, explain the general unfamiliarity of climate security with the lack of (apparently Japan-specific) research, saying: “It is also unfortunate that climate security remains unfamiliar in Japan due to a lack of research in this area” (p. 2) However, it can also be questioned whether research really has such a strong role in enabling policy discussion. At least, this does not seem like an adequate argument for using research as the material for a study on policy-making.
The authors also repeatedly express normative views about the progress of climate security, such as in the previous example: “It is also unfortunate that climate security remains unfamiliar in Japan due to a lack of research in this area” (p. 2) . However, the paper does not engage in a deeper discussion as to whether or why an increase in climate security discussion or policy is always good. Such an argument can well be made, but the authors seem to take this for granted without explicitly stating their reasoning for it.
In addition, the authors present several passing claims that remain vague and confusing at worst. For instance, according to them: “ The World Commission on Environment and Development has discussed the issue of threats and argues that one environmental threat that arises from the possibility that global warming caused by ozone layer depletion is now a global threat in the form of climate change [43]. This perspective later formed the basis of climate security theory [28].” (p.9) While this is a reference to historical literature, no reference is made to the fact that ozone layer depletion is no longer considered a major cause for climate change. It is also highly debatable whether the study that is referred to can be seen as the basis of climate security theory.
The paper offers relatively little of interest to an international audience outside the Japanese research/policy community. While the authors very well explain and justify the focus on Japan, and case studies outside Europe and the US certainly are warranted, the article appears to mainly be intended to readers already familiar with the Japanese discussion. This impression is strengthened by the fact that the authors do not open up many Japanese terms and institutions.
As the material and method are quite fixed, it is difficult to propose ways in which the paper could be improved so that its analytical contribution would be stronger. This is the main reason for the recommendation to reject the paper. However, I would suggest publishing the paper as an essay, either in this journal or somewhere else.
Reviewer 2 Report
This paper is good to publish to identify what Japan doing on climate action (SGS-13). But, it still has a limitation on reference and how to conclude this article.
Reviewer 3 Report
Specific comments:
p. 3, li. 125 Why the authors used quotes for “climate security” discourse?
p. 5, li. 204 Add a reference after Behnassi
p. 6, li. 229 'Ten' instead of '10'
p. 6, Tab. 1 English or American?
p. 7, li. 263-264 Add a reference [15]
p. 7, li. 276 Reference should be added
p. 10, li. 415-424 Not relevant, remove
p. 12, li. 555 [50] instead of [49]
p. 12, li. 560 [51] instead of [50]
p. 13, li. 578 [49–51] Check it
p. 13, li. 584 [53-55] instead of [52-54]
p. 13, li. 612 [56] instead of [55]
p. 14, li. 644 [57] instead of [56]
p. 14, li. 663, 669 [58] instead of [57]
p. 14, li. 675 [59] instead of [58]
p. 14, li. 677 [60] instead of [59]
Check all numbers of references until the end of the manuscript
p. 15, li. 733 Add a reference [65]
p. 16, li. 738 [65] instead of [63]
p. 16, li. 744 [66] instead of [64]
p. 16, li. 754 [67,68] instead of [65,66]
Check all numbers of references until the end of the manuscript
Round 2
Reviewer 1 Report
The manuscript has been adequately revised, so I recommend publishing it
Reviewer 3 Report
The authors of the paper "The Review of Climate Security Discussions in Japan" responded to all questions and suggestions. Now, the paper can be accepted for publication in International Journal of Environmental Research and Public Health.